# Airway response to respiratory syncytial virus has incidental antibacterial effects

Charles J. Sande [1,2,6], James M. Njunge [1,6], Joyce Mwongeli Ngoi[1], Martin N. Mutunga[1], Timothy Chege[1], Elijah T. Gicheru[1], Elizabeth M. Gardiner[1], Agnes Gwela[1], Christopher A. Green[2], Simon B. Drysdale[2], James A. Berkley [1,3,4], D. James Nokes [1,5] & Andrew J. Pollard[2]

RSV infection is typically associated with secondary bacterial infection. We hypothesise that the local airway immune response to RSV has incidental antibacterial effects. Using coordinated proteomics and metagenomics analysis we simultaneously analysed the microbiota and proteomes of the upper airway and determined direct antibacterial activity in airway secretions of RSV-infected children. Here, we report that the airway abundance of *Streptococcus* was higher in samples collected at the time of RSV infection compared with samples collected one month later. RSV infection is associated with neutrophil influx into the airway and degranulation and is marked by overexpression of proteins with known antibacterial activity including BPI, EPX, MPO and AZU1. Airway secretions of children infected with RSV, have significantly greater antibacterial activity compared to RSV-negative controls. This RSV-associated, neutrophil-mediated antibacterial response in the airway appears to act as a regulatory mechanism that modulates bacterial growth in the airways of RSV-infected children.

[1] KEMRI-Wellcome Trust Research Programme, Bofa Rd, Kilifi - P.O. Box 230 - 80108, Kenya. [2] Oxford Vaccine Group, University of Oxford, and the NIHR Oxford Biomedical Research Centre, Oxford, Oxford OX3 7LE, UK. [3] Centre for Tropical Medicine and Global Health, Nuffield Department of Medicine, University of Oxford, OX3 7FZ Oxford, UK. [4] The Childhood Acute Illness & Nutrition (CHAIN) Network, Nairobi - P.O. Box 43640-00100, Kenya. [5] School of Life Sciences and Zeeman Institute (SBIDER), University of Warwick, CV4 7AL Coventry, UK. [6]These authors contributed equally: Charles J. Sande, James M. Njunge. Correspondence and requests for materials should be addressed to C.J.S. (email: csande@kemri-wellcome.org)

 1

Respiratory tract infections are a major cause of morbidity and mortality globally[1,2]. In 2015, acute lower respiratory infections (ALRI) due to respiratory syncytial virus (RSV) resulted in an estimated 33.1 million disease episodes, 3.2 million hospitalisations and 118,200 deaths in children younger than the age of 5 years[3]. A common complication of respiratory virus infection is secondary bacterial infection; viruses such as influenza, RSV and parainfluenza often predispose the host to secondary respiratory disease caused by pathobionts, such as *Streptococcus pneumoniae*, *Haemophilus influenzae* and *Staphylococcus aureus*[4–8], leading to serious, potentially life-threatening sequelae[9]. Secondary bacterial complications are thought to have been the major cause of the fatalities that occurred in the wake of the major influenza pandemics of the 20th and early 21st century[8,10,11]. In the case of RSV, there is a strong link between virus infection and secondary bacterial infections, particularly otitis media caused by *S. pneumoniae* and *H. influenzae*[12–14]. Although the molecular and cellular basis for this co-pathogenesis is not fully understood, evidence from in vitro studies suggests that RSV infection of respiratory epithelial cell lines, such as HEp-2 and A549 facilitates increased adhesion of bacteria such as *S. pneumoniae* and *H influenzae*[15–18], and suggests that RSV infection in vivo increases the capacity of these bacteria to colonise the airway. These observations are supported by results from clinical studies which show that the upper airway microbiota of children with RSV infection is significantly enriched for *H. influenzae* and *Streptococcus*[19,20]. The host response to RSV infection is characterised by the recruitment of innate immune cells[21] and the release of inflammatory cytokines, such as IL-17A and other soluble mediators[22]. To date, no studies have examined whether the immune response triggered by replication of RSV in the airway has simultaneous antibacterial activity, aimed at controlling increased bacterial colonisation resulting from viral infection. Here we report the simultaneous analysis of the upper airway proteomes and microbiota of infants and children with and without RSV infection to determine whether elements of the innate immune response to RSV infection in the human airway has incidental antibacterial activity. Using shotgun proteomics, we find that the local airway response that followed RSV infection is characterised by a strong neutrophil response, which has direct antibacterial properties.

## Results

**Characteristics of the study population.** Nasopharyngeal and oropharyngeal swabs obtained from 84 children with ($n = 40$) and without ($n = 44$) RSV infection were included in the analysis. Baseline characteristics of the study population are described in Table 1. In the RSV-positive group, only children with a positive laboratory diagnosis of RSV and negative for other common respiratory viruses and *Mycoplasma pneumoniae* were included in the analysis, while the RSV-negative group comprised children who were negative for RSV and other infectious causes of respiratory illness (see "Methods" for further details). The groups were well-matched and no statistically significant differences in clinical and anthropometric measures were identified. Only

children to whom antibiotics had not been administered in the 2 weeks prior to sampling were recruited.

**The upper airway is dominated by *Streptoccocal* colonisation.** Sequence analysis of the V3–V4 regions of the 16S rRNA bacterial gene generated a total of 6,622,832 high-quality reads (median 83,831 IQR 39,686–107,741). The top 10 taxa in the airway microbiome by median proportional abundance were *Streptococcus*, *Veillonella*, *Haemophilus*, *Prevotella*, *Neisseria*, *Escherichia*, *Staphylococcus*, *Corynebacterium*, *Moraxella*, and *Leptotrichia* (Fig. 1a). *Streptococcus* was the most abundant taxon with a median proportional abundance of ~40% (Supplementary Fig. 1). Simultaneous bacterial metaproteome analysis using mass spectrometry showed a similar profile of microbial diversity and was equally dominated by Streptococcal proteins (Fig. 1b). Most of the bacterial proteins in the metaproteome were ribosomal proteins, including 50S and 30S proteins as well proteins involved in energy metabolism, such as pyruvate oxidase and enolase. Further analysis of the genomic microbiota data was undertaken to determine whether the airway replication of RSV had an effect on the most abundant bacterial taxon (*Streptococcus*). Changes in *Streptococcus* abundance during and after RSV infection were calculated in a subset of RSV-infected children ($n = 10$) for whom nasal samples were collected at admission (acute samples) and ~1 month later (convalescent samples). For 70% of these children (7/10), the airway abundance of *Streptococcus* during acute infection was higher than that at convalescence, while for three children (3/10), *Streptococcus* abundance at convalescence was either not different or had increased, relative to the acute infection time point (Fig. 1c).

**RSV infection results in airway neutrophil degranulation.** In order to characterise biological pathways that are upregulated in the local airway following RSV infection, the mean expression levels of airway proteins were compared between children with and without RSV infection. Of the 1875 host proteins that were quantified in the upper airway, the mean expression levels of 123 proteins were significantly different between the two groups of children (Supplementary data 2). Among these, 73 proteins were expressed at significantly higher levels in children with RSV infection while the remainder were higher among those without RSV infection (Fig. 2a). Gene ontology biological process (GOBP)-based enrichment analysis of these differentially expressed proteins showed that the neutrophil degranulation pathway (GOBP ID: 0043312, combined enrichment score = 401.9) was the most significantly enriched biological process among RSV-infected children (adjusted Fishers exact test $P = 9.5 \times 10^{-31}$, Fig. 2b). Other pathways such as proteolysis were also significantly enriched in the airways of RSV-infected children but their respective enrichment scores were much lower than the neutrophil degranulation pathway (combined enrichment scores of ≤ 35.1) (Fig. 2b). The neutrophil degranulation pathway comprised 33 proteins—including bactericidal proteins that are typically localised within neutrophil granules, such as MPO,

**Table 1 Comparison of clinical and anthropometric characteristics of study participants**

|  | RSV-negative ($n = 44$) | RSV-positive ($n = 40$) | *P*-value |
|---|---|---|---|
| Age (months) | 3.0 (0.03–51.3) | 3.9 (0.4–45.8) | 0.7 |
| Weight (kg) | 4.6 (1.8–14.2) | 5.6 (2.0–13.6) | 0.15 |
| Length (cm) | 61.0 (44.0–99.2) | 61.0 (47.0–95.0) | 0.2 |
| Oxygen saturation (%) | 98 (57–100) | 93 (67–100) | 0.07 |
| Respiratory rate (breaths per minute) | 56 (30–99) | 53 (11–85) | 0.6 |
| Axillary temperature (°C) | 37.7 (34.0–40.2) | 37.2 (35.7–39.6) | 0.9 |

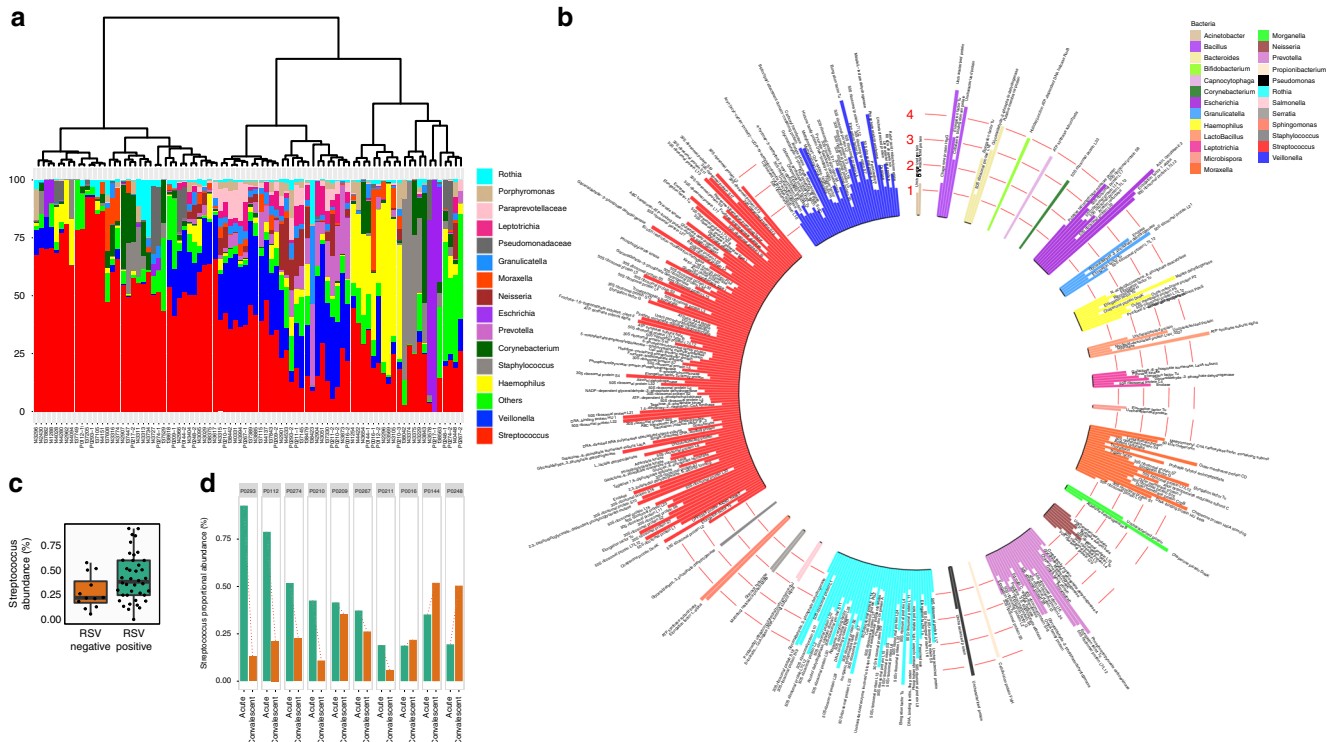

**Fig. 1** The composition of the airway microbiota. **a** Dendrogram visualising the hierarchical clustering of individual microbiomes of 84 individual children on the basis of the Bray–Curtis dissimilarity matrix. The relative abundance of the 15 highest-ranked operational taxonomic units are presented for each study participant as stacked bar charts. operational taxonomic units under the 'others' heading are an aggregate of bacteria that occurred at low abundances (<1%). The taxonomic identities of these low-abundance operational taxonomic units are presented in Supplementary Data 1. **b** The proteome of the upper airway microbiota (metaproteome) was characterised by mass spectrometric analysis using airway samples from 84 children and the results are presented in a circular bar chart. Each bar represents the mean expression level of one bacterial protein. Values on the y-axis correspond to MS protein expression levels ($\log_{10}$ reporter corrected intensity). Most of the bacterial operational taxonomic units that were identified in metagenomic analyses of the airway microbiota were also identified in the proteome. The highest number of protein identifications belonged to the genus *Streptococcus* and comprised mainly of ribosomal proteins and proteins involved in energy metabolism pathways. **c** The abundance of Streptococcus in the airway was compared between RSV-positive and RSV-negative children. RSV-infected children had a higher abundance of airway Streptococcus compared to RSV-negative children. **d** The effect of RSV infection on Streptococcal colonisation of the upper airway was characterised using a subset of 10 children from whom nasopharyngeal and oropharyngeal swabs were obtained during acute RSV infection and at convalescence, approximately one month later. Seven out of 10 children (70%) exhibited declines in the airway abundance of *Streptococcus* upon recovery from RSV, while for three children, convalescent-stage *Streptococcus* abundance was either unchanged from the acute time point or was higher. On the box and whisker plot, the bottom line on the box denotes the 25th data percentile (quartile 1), the middle line denotes the median and the upper line denotes the 75th data percentile (quartile 3). The bottom whisker represents data in the lower extremity of the distribution (quartile 1–1.5 × interquartile range) and upper whisker denotes the upper extremity of the distribution (quartile 3 + 1.5 × interquartile range). Source data are provided as a Source Data file

AZU1, BPI, LCN2, and others (Fig. 2d). In addition to neutrophil granule proteins, we also observed elevated expression of other potent antibacterial proteins in RSV-infected children; the mean expression levels of eosinophil peroxidase (EPX) and Ras-related protein (RAP1B) were significantly higher in RSV-infected children than in RSV-negative controls (Fig. 2d). Among children without RSV infection, SRP-dependent co-translational protein targeting to membrane, signal sequence recognition (GOBP ID: 0006617, adjusted Fishers exact test $P = 1.02 \times 10^{-8}$, Fig. 2c) was the most significantly enriched pathway.

**Secretions from infected children inhibit microbial growth.** Due to the significant overexpression of antibacterial proteins in the airway secretions of RSV-infected children, we reasoned that these children would have substantially greater antibacterial activity compared with children without RSV infection. To test this hypothesis, we developed an in vitro assay to measure the bacterial inhibition activity against a common nasopharyngeal pathobiont, *S. pneumoniae*. Anti-pneumococcal activity was tested in nasal samples from children with and without RSV infection and data presented as a bacterial inhibition index (BII), whose values ranged from 0 (no inhibition) to 40,960 (complete inhibition) (Fig. 3a). Children were binned into five BII strata (BII: ≤20, 40, 80, 160 and ≥320) and the relative proportions of different OTUs was determined within each group. The proportional abundance of *Streptococcus* decreased with increasing BII, with children who had the lowest anti-pneumococcal activity (BII: ≤20) having the highest mean airway abundance of *Streptococcus* (45%) while those with the highest anti-pneumococcal activity (BII ≥ 320) had the lowest *Streptococcus* abundance (25%) (Fig. 3b and Supplementary Fig. 4).

We then analysed the correlation between the anti-pneumococcal activity measured in the bacterial inhibition assay and the expression levels of 15 neutrophil granule proteins that were significantly overexpressed in children with RSV infection (highlighted in Fig. 3d). We stratified this correlation analysis by neutrophil granule subset and included two representative members of each subset (MPO and AZU1 for azurophilic granules, LCN2 and ITGAM for specific granules, and MMP9

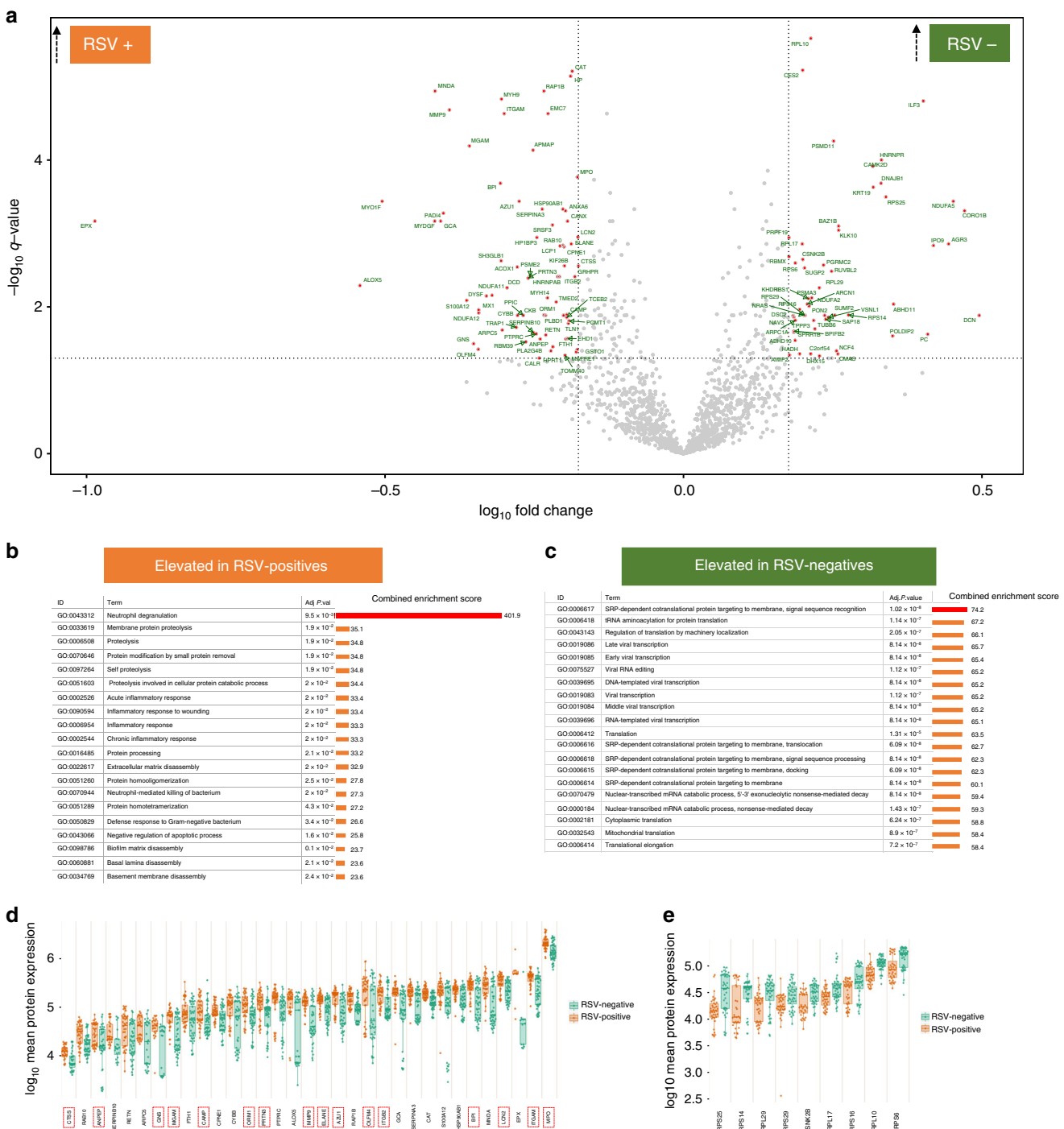

**Fig. 2** Analysis of the airway proteome by tandem mass spectrometry. Analysis of airway proteomes was undertaken using mass spectrometry in a population of 40 children with RSV infection and 44 RSV-negative children. **a** Differences in the mean expression levels of local airway proteins between the RSV-positive and RSV-negative children was analysed using two-sided Student's t-tests and presented in a volcano plot. The y-axis represents $-\log_{10}$ fdr-adjusted P values (q values) computed using two-sided Student's t-tests and the x-axis represents $-\log_{10}$ fold change in protein expression. Red circles represent differentially expressed proteins (defined by q values: ≤0.05 and ≥1.5-fold change in expression between groups). Red circles in the top left and top right quadrants represent proteins that were significantly elevated in RSV-positive and RSV-negative children, respectively. **b**, **c** Gene ontology (GO)-based biological pathways analysis was used to identify biological processes that were significantly overrepresented in RSV-positive and RSV-negative children. Those that were significantly overrepresented in RSV-positive children are shown in **b** while those that were overrepresented in RSV-negative children are shown in **c**. Adjusted P-values in **b** and **c** were calculated on the enrichR platform and are based on the Fishers extract test. The most significantly enriched pathway in RSV-positive children was neutrophil degranulation and in RSV-negative children was SRP-dependent co-translational protein targeting to membrane, signal sequence recognition. **d**, **e** The specific proteins that were significantly differentially expressed within those pathways are shown in **d** and **e**, respectively. The proteins highlighted with the red boxes in **d** are neutrophil granule proteins. The elements of the box and whisker plots are defined in Fig. 1. Source data are provided as a Source Data file

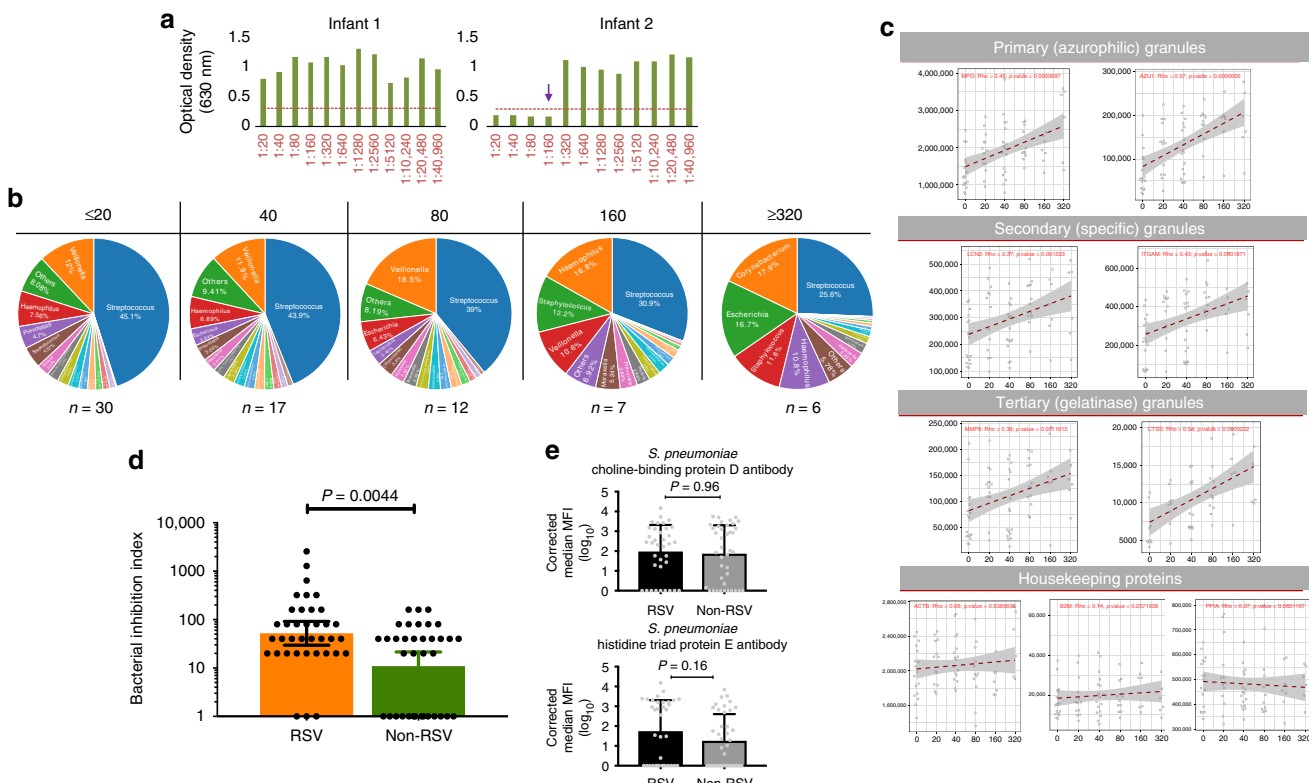

**Fig. 3** Analysis of bacterial inhibition activity in airway secretions. **a** Example plots of children with different bacterial inhibition activities. The *y*-axis represents the optical density which is directly related to the amount of bacterial (*Streptococcus pneumoniae*) growth: bacterial growth increases turbidity of the culture media resulting in a proportional increase in optical density. The *x*-axis represents a doubling dilution of nasal samples from 1:20 to 1:40,960. The dashed horizontal red line denotes an optical density cutoff above which substantial turbidity of the culture media was observed. Nasal secretions from infant 1 were unable to inhibit bacterial growth at any dilution (bacterial inhibition index [BII] = 0). Nasal secretions from infant 2 completely inhibited pneumococcal growth at dilutions ≤ 1:160 (BII = 160). **b** Children were categorised into five BII strata and differences in the proportional abundance of different bacterial taxa in each stratum presented in pie charts. The airway abundance of *Streptococcus* decreased with increasing bacterial inhibition activity. **c** Bacterial inhibition activity in nasal samples (*N* = 72) was correlated with the expression levels of neutrophil-granule proteins (stratified by functional subset). Representative proteins from all three subsets of neutrophil granules (azurophilic, specific and gelatinase) were positively correlated with bacterial inhibition activity (complete analysis of the correlation between bacterial inhibition activity and all the neutrophil granule proteins can be found in Supplementary Fig. 2). In contrast, there was no correlation between the expression levels of housekeeping proteins (ACTB, B2M and PPIA) and bacterial inhibition activity in nasal samples. Correlation analysis was done using Spearman's rank order correlation. **d** Bacterial inhibition activity was compared between RSV-positive (*N* = 36) and RSV-negative children (*N* = 36). The mean BII level in RSV-infected children (geometric mean: 52) was significantly greater than the mean BII in RSV-negative children (geometric mean: 10.9). Error bars indicate 95% confidence intervals. **e** Mean airway IgG levels to two *Streptococcus pneumoniae* proteins (choline-binding protein D and Histidine Triad protein E) were compared between RSV-positive (*N* = 36) and RSV-negative children (*N* = 36). Error bars indicate 95% confidence intervals. *P*-values in **d** and **e** were based on Mann–Whitney *U*-test. Source data are provided as a Source Data file

and CTSS for gelatinase granules) (Fig. 3c). We observed a significant positive correlation between the expression level of all neutrophil granule proteins and pneumococcal inhibition activity (Fig. 3c and Supplementary Fig. 2). Expression levels of three housekeeping proteins: beta-actin (ACTB), beta-2-microglobulin (B2M) and peptidylprolyl isomerase A (PPIA) were used as controls and their respective expression levels did not correlate with bacterial inhibition activity (Fig. 3c).

We then compared pneumococcal inhibition activity between RSV-positive and RSV-negative children. Airway secretions of RSV-infected children had significantly greater bacterial inhibition activity (BII geometric mean: 52) compared to RSV-negative children (BII geometric mean: 10.9, Mann–Whitney test *P*-value = 0.004) (Fig. 3d). Further analysis of a subset of RSV-infected children for whom paired acute and convalescent samples were collected showed a similar relationship: airway secretions sampled at the time of infection, had significantly greater bacterial inhibition activity compared to samples collected a month later, during convalescence (Supplementary Fig. 3). To determine

whether this difference could be explained by differences in the mean levels of pneumococcus-specific antibody between RSV-positive and RSV-negative children, we measured the levels of airway anti-pneumococcus IgG to two pneumococcal proteins (choline-binding protein D and Histidine triad protein E). The results of these analyses showed that the mean IgG levels to these proteins was not significantly different between RSV-positive and RSV-negative children (Fig. 3e).

**RSV infection is associated with neutrophil phagocytosis.** We used flow cytometry to characterise airway-resident neutrophils, which are the source of the neutrophil granule proteins that were identified in the proteomic analysis (see "Methods" and Supplementary Fig. 5 for a full description of the FACS analysis and gating strategy). We also used flow cytometry to assess the functional activity of these cells in RSV-positive and RSV-negative children. In RSV-infected children, neutrophils occurred at a median frequency of 12% while in RSV-negative children,

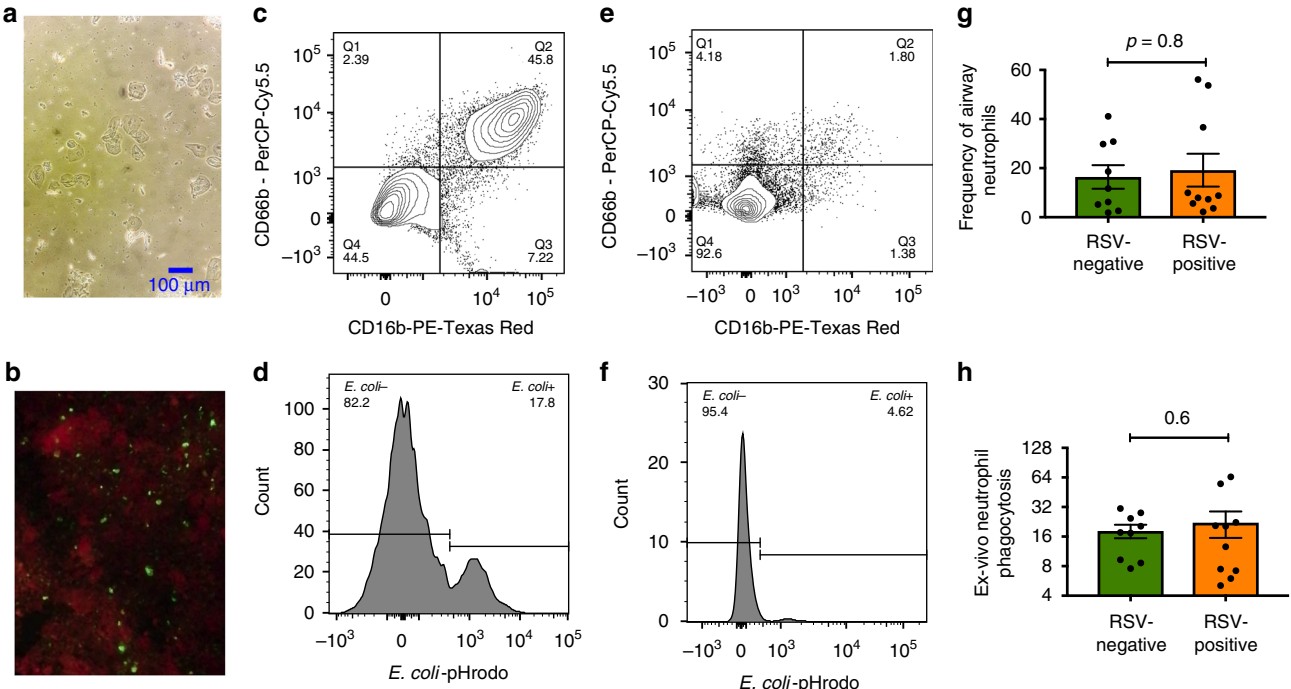

**Fig. 4** Analysis of the cellular sources of the airway proteome by flow cytometry. **a** Cells obtained from nasopharyngeal and oropharyngeal swabs (both eluted in universal transport media) from a child with an RSV infection was visualised at ×200 magnification. Multiple cell morphologies including epithelial cells were observed. **b** Upper airway sample obtained from an infant admitted to hospital with RSV-pneumonia was analysed using a direct immunofluorescent antibody test[38] (Light Diagnostics, Merc). RSV-infected cells are shown emitting an apple-green fluorescent signal, while uninfected cells emit a red fluorescent signal from a counterstain (Evans blue). Image taken using Nikon Eclipse fluorescent microscope image acquisition software. **c** Flow cytometry analysis was used to identify neutrophils in the sample described in **b** above (details of full gating strategy can be found in the "Methods" section and in Fig. S5). Cells that were double positive for CD66b and CD16 were gated as neutrophils. **d** Phagocytic activity of airway-resident neutrophils shown in **c** above was analysed using flow cytometry. Cells were co-incubated with an *E. coli* strain that was labelled with a dye (pHrodo) whose florescence was only activated upon phagocytosis. In this example, 17.8% of the neutrophils in the sample exhibited bacterial phagocytosis. **e**, **f** A similar example is shown from a different child who had fewer neutrophils and also had reduced phagocytic activity. **g** The frequency of airway-resident neutrophils in RSV-positive and RSV-negative children is shown as a proportion of all airway cells. The proportion of airway cells that were identified as neutrophils was greater in RSV-positive children ($N = 10$) compared to RSV-negative children ($N = 10$) although the difference did not reach statistical significance. Error bars indicate 95% confidence intervals about the geometric mean. **h** The capacity of airway neutrophils from RSV-positive ($N = 10$) and RSV-negative ($N = 10$) children to phagocytose bacteria was compared. A greater proportion of neutrophils from RSV-positive children exhibited phagocytic activity compared to RSV-negative children, although this difference did not achieve statistical significance. Error bars indicate 95% confidence intervals about the geometric mean. Comparison between the RSV-positive and RSV-negative groups in **g** and **h** was conducted using the Mann–Whitney *U*-test. Source data for **g** and **h** are provided as a Source Data file

neutrophils comprised 8% of cells in the upper airway (Fig. 4g). Although the mean frequency of airway-resident neutrophils in RSV-positive children was higher than that of RSV-negative children, this difference was not statistically significant. Next, using an ex-vivo phagocytosis assay, we assessed the ability of airway-resident neutrophils from RSV-positive and RSV-negative bacteria to ingest and kill bacteria. In this assay, we used a representative bacteria species (*E. coli*) that was labelled with a tag (pHrodo) whose fluorescence was activated only upon ingestion and localisation of the bacteria into a phagosome, thereby allowing for a quantitative assessment of phagocytic activity through flow cytometry (see examples in Fig. 4d, f). Using this assay we compared the phagocytic capacity of airway neutrophils obtained from RSV-positive and RSV-negative children. Although not statistically significant, a greater proportion of airway neutrophils obtained from RSV-positive children exhibited phagocytic activity against *E. coli* (median 18%) compared to RSV-negative children (median 16%).

**Degranulation is associated with lower oxygen saturation**. In order to identify a potential role for airway neutrophil granule

proteins in the pathogenesis of RSV, we examined the effect of different protein expression levels on clinical parameters associated with RSV disease severity. Since one of the central features of severe RSV pneumonia is reduced blood oxygen saturation resulting from occlusion of small airways with necrotic, virus-infected cells as well as infiltrating immune cells, we therefore examined the effect of the increased expression of neutrophil granule proteins on blood oxygen saturation. Children were dichotomised into two groups (high and low neutrophil granule expressers) based on the median expression level of different proteins. Children whose protein expression level was above the median expression (of the full study population) were classified as high expressers while those who were below the median were classified as low expressers (Fig. 5). We then compared oxygen saturation levels between these two groups. For all neutrophil granule proteins that were analysed (ITGAM, BPI, LCN2, MGAM, ITGB2, AZU1, ANPEP, MMP9 and MPO), children with low levels of airway neutrophil granule expression had significantly higher blood oxygen saturation levels compared to children in the high-expression group (Fig. 5a). For comparison, we also dichotomised the expression levels of two house-keeping proteins—beta-actin (ACTB) and β-2-microglobulin

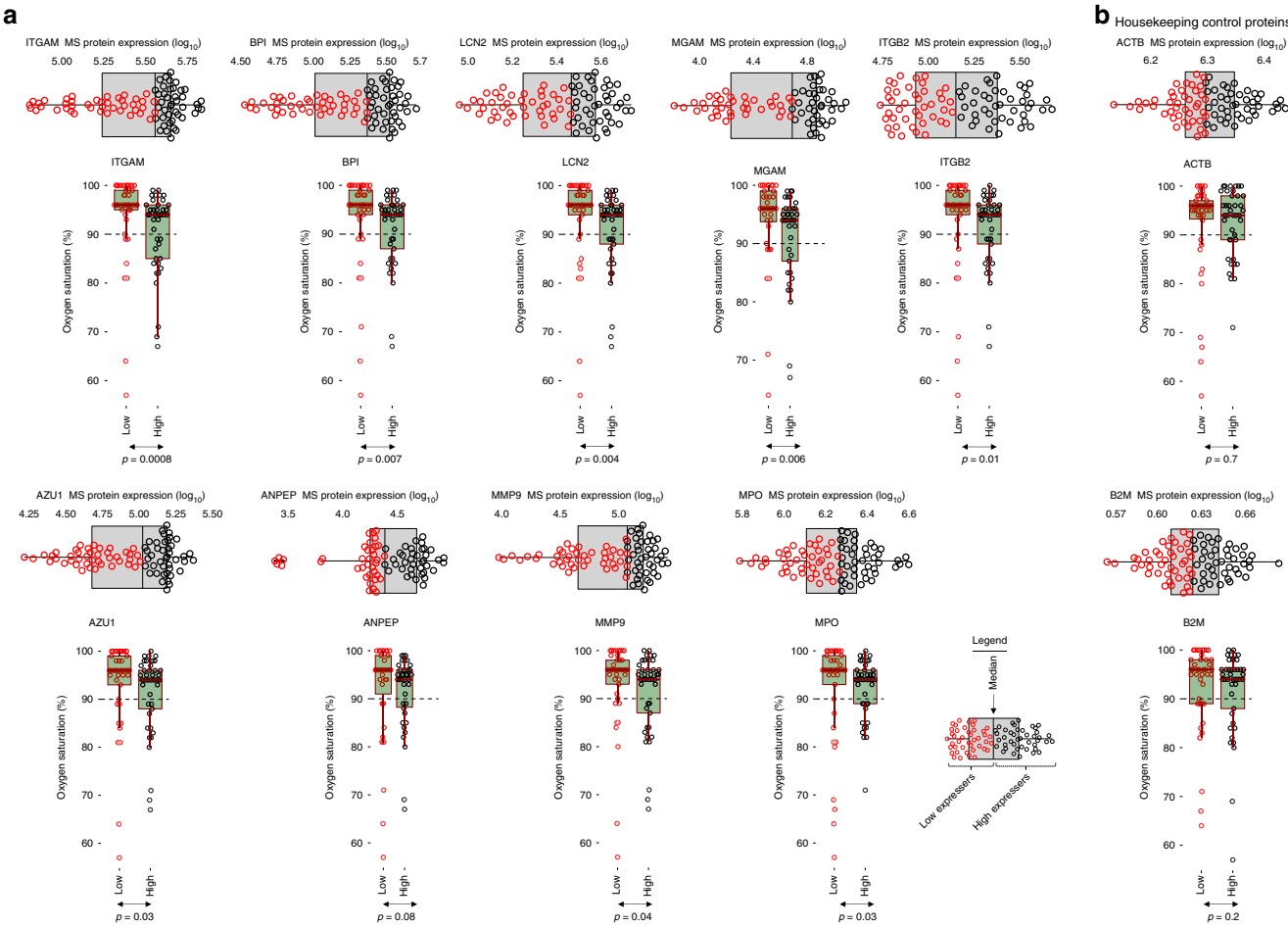

**Fig. 5** Association between airway neutrophil granule expression and disease severity. In order to characterise the clinical implications of the increased expression of neutrophil-associated proteins on disease pathology, we assessed the association between the level of expression of these proteins and a cardinal feature of RSV pathology, blood oxygen saturation. **a** Using proteome data obtained from all admitted children ($n = 84$), we stratified the expression levels of nine neutrophil granule proteins (ITGAM, BPI, LCN2, MGAM, ITGB2, AZU1, ANPEP, MMP9 and MPO) that were significantly over-expressed in RSV-infected children by the median. Children whose protein expression level was above the population median were classified as high-expressers while those below it were classified as low expressers (see legend for graphical illustration of the stratification strategy). For all proteins analysed, children in the high expression group had significantly lower blood oxygen saturation levels (measured using fingertip pulse oximetry) compared to children in the low expression group. **b** For comparison, we used the same median stratification strategy to dichotomise children into high and low expression groups for two housekeeping control proteins, beta-actin (ACTB) and beta-2-microglobulin (B2M). For both proteins there was no statistically significant difference in blood oxygen saturation levels between high and low expressers. The dashed horizontal line indicates an oxygen saturation threshold of 90%, below which children would typically be characterised as functionally hypoxic. Statistical comparison between high and low strata was done using a two-sided Student $t$-test. The elements of the box and whisker plots are defined in Fig. 1. Source data are provided as a Source Data file

(B2M)—using the median. In contrast to the neutrophil granule proteins, there was no difference in blood oxygen saturation between high and low expressers of ACTB and B2M (Fig. 5b).

## Discussion

RSV infection in infants and children has been associated with an increased risk of secondary bacterial infections, such as otitis media mainly due to secondary *Streptococcus* infection[12–14]. We therefore hypothesised that the host response to RSV is partly directed at controlling increased levels of *Streptococcus*. We undertook comprehensive analysis of the airway microbiota and proteomes of children with RSV as well as RSV-negative controls, in order to identify features of the post-infection innate immune response that have incidental antibacterial effects. We found that the upper airways of children in the study was dominated by a highly diverse population of commensal bacteria for which a wide range of bacterial proteins could be detected in the airway

metaproteome. We also found evidence that infection with RSV is associated with increased Streptococcal colonisation. In a subset analysis of children who were sampled both during acute disease and at convalescence, approximately one month later, we found that the airway abundance of *Streptococcus* was greater at the time of RSV infection compared with the convalescent time point when children had cleared the infection. These results agree with previous clinical studies that have shown that children with RSV infections have a higher abundance of *Streptococci* in their airway secretions[19], and supports the idea that RSV-infection in the human airway, enhances Streptococcal colonisation.

Our results showed that following an RSV infection there was a significant increase in the expression levels of a large number of proteins related to innate immunity. Due to the fact that RSV replication and the RSV-mediated increase in bacterial colonisation occur simultaneously, it was not possible to establish which elements of this response were directly attributable to the virus and which ones were triggered by the increased bacterial load.

Since the innate host response to infection is highly co-ordinated, and the proteins that were highly expressed in RSV-positive children are likely to represent functionally related proteins that are co-regulated in order to initiate an effector response, we used pathway enrichment analysis to identify the innate immune pathways that were associated with the highly expressed proteins. In RSV-infected children, we found a significant enrichment of the neutrophil degranulation pathway, characterised by the increased expression of neutrophil effector proteins, such as MPO, AZU1 and ELANE which have been shown in previous work to be associated with RSV infection[23–25]. In agreement with previous studies[26,27], we found that neutrophils could be detected at relatively high frequencies in the upper airway samples of children with respiratory infections, thereby suggesting that the local innate response to RSV infection in the airway is associated with an influx of neutrophils and their subsequent degranulation. Neutrophils are specialised innate phagocytic cells that are densely packed with granules containing hundreds of proteins with broad antibacterial action[28]. These granules can be broadly classified into three subsets[29]: primary (or azurophil) granules (characterised by expression of MPO, BPI and ELANE), secondary (or specific) granules (characterised by LCN2 expression) and tertiary (or gelatinase) granules (identified by expression of MMP9). In this study, we identified proteins belonging to all three subsets in the upper airway secretions of study participants and all were expressed at significantly higher levels in RSV-infected children compared to RSV-negative controls. Previous studies have provided compelling evidence of the direct anti-bacterial action of neutrophil granule proteins and have high-lighted their central role in innate immune defence against bacteria. For example, MPO is a potent bactericidal protein, which in the presence of a hydrogen peroxide ($H_2O_2$) source—such as *S. pneumoniae*[30]—selectively binds to bacteria, such as *Escherichia coli*, *Pseudomonas aeruginosa*, *S. aureus* and *Streptococcus pyogenes*, causing their rapid killing[31]. BPI on the other hand exhibits independent antibacterial activity against Gram-negative bacteria such as *E. coli*[32,33] and is also capable of neutralising bacterial endotoxin[34], while ELANE mediates cleavage of virulence factors from bacteria such as *Salmonella* and *Shigella*[35]. In addition to neutrophil granule proteins, we found increased expression of bactericidal proteins from other innate immune cell populations, such as the eosinophilic peroxidase EPX, which rapidly kills bacteria such as *Mycobacterium tuberculosis*[36]. These observations suggested that the neutrophil response that follows an RSV airway infection might have significant antibacterial activity due to the incidental overexpression of these bactericidal proteins. To test this hypothesis, we compared the relative bacterial inhibition activity in airway samples obtained from RSV-positive and RSV-negative children, as well as a subset of RSV-infected infants from whom acute and convalescent samples were available. We found that the airway secretions from children infected with RSV had significantly higher bacterial inhibition activity compared with those from RSV-negative children. Additionally, bacterial inhibition activity at the acute stage of RSV infection, was significantly greater than during convalescence. Furthermore, there was a positive correlation between the expression levels of neutrophil granule proteins and inhibition of pneumococcal growth in the in vitro bacterial inhibition assay. We also found that an increase in bacterial inhibition activity was associated with changes in the composition of the upper airway microbiota, characterised by a reduction in the airway abundance of *Streptococcus*. Comparative analysis of the relative levels of anti-pneumococcus IgG in RSV-positive and RSV-negative children showed that the difference in bacterial inhibition activity could not be explained by a differential abundance of these antibodies, since antibody levels against two different

pneumococcal antigens were not significantly different. Taken together these observations suggest that the recruitment and degranulation of airway neutrophils in following an RSV infection is associated with the increased expression of bactericidal proteins whose antibacterial action contributes to the regulation of commensal bacteria in the upper respiratory tract. Due to the fact that the virus-mediated increase in bacterial load occurs concurrently with the airway replication of RSV, it was not possible to conclusively establish whether the antibacterial activity that we observed in RSV-infected children was as a result of virus infection, increased bacterial load or both. Despite the fact that nasal secretions from RSV-infected children had significantly greater antibacterial activity compared to RSV-negative controls, these children had a higher airway abundance of *Streptococcus*. A possible explanation for this observation is the contemporaneous nature of RSV replication and increased bacterial colonisation in the airway. Previous studies in in vitro cell culture models have shown that *S. pneumoniae* adheres at significantly greater levels to RSV-infected cells, compared to uninfected cells and that this increased adherence is mediated by a direct interaction between bacterial adhesins and the RSV attachment protein that is expressed on the surface of RSV-infected cells[15,16,18]. This suggests that the conditions in the airway at the time of acute RSV infection—when virus replication is at peak—favour increased bacterial colonisation and that these conditions persist for as long as the viral infection remains unresolved.

In order to characterise the clinical implications of neutrophil-mediated inflammation and degranulation on the pathology of RSV disease, we examined the association between the expression level of neutrophil granule proteins and a key feature of RSV disease pathology, reduced blood oxygen saturation. For all the proteins that we analysed, we found that increased expression of neutrophil granule proteins in the airway was associated with reduced oxygen saturation. These observations suggest that the influx of neutrophils in the airway and their subsequent degranulation may play a substantial role in the pathology of RSV disease. Despite over 60 years of research, remarkably little is known about the cellular response to RSV in the lungs of infants. The most detailed study to date examined archived post-mortem lung tissue from suspected fatal cases of RSV bronchiolitis from the 1930s and 1940s. Lung tissue obtained from these infants was characterised by a neutrophil influx that was distributed predominantly between arterioles and airways[37]. Using modern systems biology tools, our results of our study align with these previous findings and highlight a potential central role of neutrophils in RSV disease pathology. It is likely that these infiltrating neutrophils occlude the small airways of young infants, leading to impaired gas exchange and manifesting clinically in reduced blood oxygen saturation and hypoxia.

Using flow cytometry analysis we found that airway samples obtained from RSV-infected children had a greater abundance of neutrophils compared to those from RSV-negative children, although this difference did not reach statistical significance. We also found that a greater proportion of neutrophils from RSV-infected children exhibited phagocytic activity against bacteria, although the difference was also not significant. A potential explanation of these results is the relatively small sample size used in the cellular analysis.

A further limitation of this study is the fact that the interactions between the host, virus and resident microbiota were studied in the upper airway. Since the pathogenesis of pneumonia is limited to the lower respiratory tract, future studies should aim to examine these interactions using samples derived from the lung. Such studies may provide a unique opportunity to examine the mechanistic basis for serious respiratory infections such as RSV.

## Methods

**Study site and population**. The study was conducted in Kilifi County, a predominantly rural community on the Kenyan coast. The study population was recruited from local healthcare facilities including the Kilifi County Hospital and the Pingilikani dispensary. Nasopharyngeal and oropharyngeal swabs were collected and used for multiplex PCR diagnosis of 15 respiratory pathogens: RSV (A and B), rhinovirus, parainfluenza virus (1, 2, 3 and 4) adenovirus, influenza (A, B and C), coronavirus (OC43 and e229), human metapneumovirus and *M. pneumoniae*. Results from RSV-positive samples were confirmed using a direct fluorescent antibody test kit (Light Diagnostics, Merc) in accordance with the manufacturer's instructions. A standardised set of clinical and anthropometric measures were collected for all study participants including: age, respiratory rate, axillary temperature, oxygen saturation, and history of upper respiratory tract symptoms (coughing, sneezing, runny nose). To establish associations between RSV infection and the resident airway microbiota, children were stratified into two groups, RSV-positive and RSV-negative: in the RSV-positive group, only children who were positive for RSV (real-time cycle threshold values ≤27) and negative for all other targets on the multiplex panel were included. In the RSV-negative group, only children who were negative for all pathogens in the real-time PCR panel were included. Children with positive blood culture results were excluded from the study. Written informed consent was sought from the parents and legal guardians of all children who were sampled in this study. Ethical approval for the conduct of this study was granted by the Kenya Medical Research Institute's Scientific and ethical research unit (SERU). All study procedures were conducted in accordance with Good Clinical Laboratory Practise (GCLP) standards. All measurements reported in this paper were obtained from individual children sampled once, with the exception of a subset of 10 children who were sampled during acute disease and at convalescence in order to examine the effect of RSV infection on Streptococcal colonisation.

**Sequencing of the bacterial 16S rRNA gene**. The swabs that were collected from each study participant were initially stored at −80 °C. For DNA extraction, samples were retrieved from storage, thawed on ice and cell pellets obtained by centrifuging samples for 10 min at $17,000 \times g$, after which they were washed once using sterile phosphate buffered saline (PBS). 350 µl of a cell lysis buffer (RLT, qiagen) supplemented with β-mercaptoethanol (1%) was added to the pellet and vortexed briefly to break up the pellet. Glass beads were then added to the sample and homogenisation was done by vortexing at maximum speed for 1 min. DNA extraction from the cell homogenate was done using the AllPrep RNA/DNA/ Protein kit (Qiagen), following the manufacturer's instructions. DNA yields were quantified using a Qubit 2.0 fluorometer (Thermo Fisher).

The V3–V4 hypervariable region of the 16S rRNA gene was targeted for sequencing. Primers targeting this region were constructed with Illumina adaptor overhang sequences added to the gene-specific primer sequences. A region of ≈550 bp was targeted and amplified using the 341F (5′-CCTACGGGNGGCWG CAG-3′) and 785R (5′-GACTACHVGGGTATCTAATCC-3′) primers appended with Illumina adaptor sequences. Amplifications were done in 25 µl reactions with 12.5 µl Q5® Hot Start High-Fidelity 2X Master Mix (NEB), 1 µl of 1 µM forward and reverse 16S Amplicon PCR primer and 2.5 µl of template. The reactions were performed on ABI Veriti thermocyclers (Applied Biosytems) under the following conditions: 95 °C for 3 min, 25 cycles of: 95 °C for 30 s, 55 °C for 30 s, 72 °C for 30 s, followed by 72 °C for 5 min and a final hold at 4 °C. The amplified products were then verified on 1% agarose gel before purifying using Agencourt AMPure XP beads (Beckman Coulter). A negative $dH_2O$ control was incorporated into the entire experimental workflow and analysed by agarose gel electrophoresis in order to ensure that the workflow was free of extraneous bacterial contamination. This control was also included in subsequent library preparation and sequencing steps and confirmed to be free of extraneous bacterial contamination.

Libraries were prepared by ligating Illumina dual indices and Illumina-sequencing adaptors to the prepared amplicons using the NexteraXT index kit. Attachment of the indices was performed using 5 µl of the 16S amplicon DNA, 5 µl of Illumina Nextera XT Index Primer 1 (N7xx), 5 µl of Nextera XT Index Primer 2 (S5xx), 25 µl of Q5® Hot Start High-Fidelity 2X Master Mix (NEB), and 10 µl of PCR-grade water (Ambion). The reactions were performed on ABI Veriti thermocyclers (Applied Biosytems) under the following conditions 95 °C for 3 min, followed by eight cycles of: 95 °C for 30 s, 55 °C for 30 s, and 72 °C for 30 s, and a final extension at 72 °C for 5 min and a final hold at 4 °C. Each library was then purified using Agencourt AMPure XP beads (Beckman Coulter), and thereafter the size distribution and library quality control were carried out using the Agilent 2100 Bioanalyzer (Agilent) to confirm the expected size distribution and the quality. The libraries were quantified using the Qubit 2.0 fluorometer (Life Technologies). The barcoded libraries were normalised and pooled at equimolar concentration based on the Qubit results and 8 pM of the pooled library spiked with 5% Phix (v3) for sequencing. Sequencing was done on the Illumina MiSeq system using $2 \times 300$ bp PE sequencing with the MiSeq® Reagent Kit v3 (600 cycles).

**Analysis of the airway microbiome**. A customised Python-based Quantitative Insights Into Microbial Ecology 2 (https://qiime2.org/)—QIIME2[39,40]—workflow was adopted for analysis of raw Illumina reads and generation of the sequence table. Sequence quality control and feature table construction was done using

DADA2[41]: low-quality sequencing reads were filtered out by trimming the reads at the nucleotide position at which a substantial drop in quality was observed. Error estimation, de-replication, construction of the sequence table and removal of chimeras was done using default DADA2 parameters. Taxonomies were assigned to the sequence table using a pre-trained Naïve Bayes classifier. The classifier was trained on the Greengenes[42] 13_8 99% OTUs in which sequences were trimmed to represent the 16S region spanning 341F/785R V3/V4 primer pair. The sequence data have been submitted to the European Nucleotide Archive database (PRJEB28053).

**Analysis of the airway proteome**. The host airway proteomes of children in this study were characterised in nasopharyngeal and oropharyngeal swabs using high-performance liquid chromatography–tandem mass spectrometry (HPLC–MS/MS)[43]. A detailed description of this method can be found in Supplementary methods. The data arising from the mass-spectrometry analysis are deposited in the ProteomeXchange Consortium database[44] (Accession number: PXD009403). Analysis of the airway metaproteome was undertaken by searching the Raw mass spectrometer files (using MaxQuant software version 1.6.0.131) against a bacterial FASTA database, which was constructed by assembling Uniprot FASTA files of individual bacteria species that had been identified by genomic microbiota analysis into a single FASTA search file. Searches for bacterial proteins were done using the Andromeda search engine[45]. To test whether the mean expression levels of different host airway proteins were significantly different between RSV-positive and RSV-negative children, two-sided *t*-tests were conducted and adjustment for multiple testing done using the false discovery rate (fdr) method. Proteins were significantly differentially expressed if their mean expression level in one group was at least 1.5-fold greater than in the alternative group. GO biological pathway enrichment analysis was done on the enrichR bioinformatic pipeline[46,47]. The output of this analysis was used to determine whether certain biological pathways were statistically overrepresented in the airway secretions of RSV-positive or RSV-negative children. The difference between the blood oxygen saturation levels of children with high (above median) neutrophil granule expression levels and those of children with low (below median) expression levels was analysed using Mann–Whitney test.

**Bacterial inhibition assay**. Using a pure overnight culture of *S. pneumoniae* 0.5 McFarland bacterial suspension was prepared in Mueller–Hinton broth with Tris ethylenediaminetetraacetic acid sodium dodecyl sulfate (TES) buffer (Thermo-fisher) and standardised using a nephelometer. 100 µl of the suspension was transferred to a tube of Mueller–Hinton broth containing lysed horse blood. 50 µl of a 1:32 dilution of this mixture in minimum essential medium (MEM; Sigma Aldrich) was then added to the wells of a U-bottomed 96-well plate. Nasal samples whose bacterial inhibition activity was to be measured were prepared as follows: 500 µl of raw nasopharyngeal and oropharyngeal swabs eluted in universal transport media (Copan diagnostics) were centrifuged at $17,000 \times g$ for 10 min and supernatants obtained. The supernatants were further spun at $17,000 \times g$ for 30 min on 3k Amicon centrifugal filter devices (Merck), to remove low molecular weight compounds (≤3 kDa). For each sample, the resulting concentrate was diluted over a 12-step doubling dilution series ranging from 1:20 to 1:40,960 in MEM. 50 µl of this preparation was then added to the plate containing the bacterial solution that had been prepared earlier and the plates were then incubated for 36 h in a 37 °C incubator. After the incubation, the plates were visually inspected by two independent readers to determine the highest sample dilution at which bacterial growth was completely inhibited. The reciprocal of this dilution was referred to as the bacterial inhibition index (BII). Plates were also read on an optical microplate reader (BioTek) and optical densities at 630 nm determined. The correlation between airway protein expression level and bacterial inhibition activity was examined using Spearman's correlation analysis while the difference in bacterial inhibition activity between RSV-positive and RSV-negative children was tested using the Wilcoxon signed-rank test.

**Analysis of *Pneumococcal* antibodies in airway secretions**. Two conserved pneumococcal antigens, pneumococcal histidine triad protein E[48] and choline-binding protein D[48] were expressed as synthetic peptides and dissolved in a solution containing 70% dimethyl sulfoxide, 5% glycerol and 25% sodium acetate (pH 4.5). Each antigen was spotted on a functionalized Epoxy glass slide using a non-contact Inkjet microarray printer (Array jet). Slides were then dried at room temperature (RT) for 48 h and subsequently loaded onto hybridisation cassettes where they were initially washed twice with PBS containing Tween-20 and then twice with PBS only. Blocking was done for 1 h using 5% bovine serum albumin (BSA) in 1× saline sodium citrate (SSC) buffer (pH 7.0) containing 0.25% sodium dodecyl sulfate (SDS) and 4.5% sodium chloride (NaCl). Blocked slides were incubated at 41 °C on a shaking platform. Nasal samples from study participants were first spun at $17,000 \times g$ for 5 min to obtain supernatants which were then filtered on 3K Amicon centrifugal filter devices (Merck). The resulting concentrates were supplemented at a 1:1 ratio with 5% BSA in PBS containing Tween-20. A total of 50 µl of this solution was added onto hybridisation cassette wells and the slides incubated for 4 h at 4 °C on a shaking platform. After incubation, slides were washed as before. 1:500 dilutions of secondary antibodies—goat anti-human IgA

conjugated to Alexa-fluor 555 (Southern biotech) and goat anti-human IgG conjugated to Alexa flour 647 (Invitrogen)—were mixed together and then added onto the slides, which were left to incubate for 1 h at 25 °C. Following the incubation, slides were washed as above, disassembled from the cassettes, rinsed using Milli-Q water and spun dry for 5 min using a slide spinner. Slides were then read on a GenePix 4300A microarray scanner (Molecular devices). Pneumococcus-specific antibody levels against the two antigens were calculated automatically by the scanner software and expressed as background-corrected median fluorescence intensities (corrected median MFI). The difference in the levels of pneumococcus IgG between RSV-positive and RSV-negative children was tested using the Wilcoxon signed-rank test.

**Flow cytometry analysis of upper airway cells.** One millilitre of nasopharyngeal and oropharyngeal swab samples obtained from children was centrifuged at 17,000×g for 7 min, after which 800 μl of the supernatant was removed and discarded. The remaining 200 μl were split into two aliquots of 100 μl each. The first aliquot was used for neutrophil phenotyping assays and the other was used for neutrophil phagocytosis assays. A volume of 20 μl of a pre-constituted cocktail of the following antibodies (from ThermoFisher) was used to label both aliquots— CD45, CD16, CD14, CD3, CD19, HLA-DR, CD66b, CD11b and a Live–dead marker. With the exception of the live/dead marker, all other antibodies were diluted 1:100 in FACS buffer. The live/dead marker was prepared at a 1:1000 dilution in FACS buffer. For the phagocytosis assay tube, 20 μl of opsonised *E. coli* was added to the tube (pHrodo Red *E. coli* BioParticles; ThermoFisher). The bacteria was initially prepared by mixing the *E.coli* strain with new-born calf sera followed by a 30-min incubation at 37 °C. After this step, both tubes were incubated at 37 °C for 35 min. After the incubation, 20 μl of a live–dead marker was added to each tube and incubated for 10 min at 37 °C. The reaction was stopped by adding 500 μl of 1X RBC lysis buffer followed by a 5 min incubation. Cells in each tube were then spun down at 2700×g for 1 min and the supernatant discarded. Cells were then washed twice with FACS buffer, after which 350 μl of FACS flow was added. Cells were then analysed immediately on a BD LSR Fortessa instrument. The following gating strategy was used to detect airway-resident neutrophils: Debris were excluded on the basis of their forward (FSC-A) and side (SSC-A) scatter characteristics (Supplementary Fig. 5a). Doublets were excluded using FSC-A versus FSC-H (Supplementary Fig. 5b) and dead cells were excluded using the live–dead marker (Supplementary Fig. 5c). Cells that were double positive for CD66b and CD16 were gated as neutrophils (Supplementary Fig. 5e). Cells in this gate were further analysed for their ability to phagocytose a representative bacteria (*E. coli*) and the results expressed in a histogram plot (Supplementary Fig. 5d). Data was analysed using FlowJo software.

**Reporting summary.** Further information on research design is available in the Nature Research Reporting Summary linked to this article.

## Data availability
Data underlying Figs. 1a, 1b, 1c, 1d, 2a, 2b, 2c, 2d, 2e, 3a, 3b, 3c, 3d, 3e, 4g, 4h, 5, and Supplementary Figs. 1, 2, 3a and 3b are provided as Source Data files. The proteomics and metagenomics data sets containing all the raw data and associated metadata underlying Figs. 1a, 1b, 1c, 1d, 2a, 2b, 2c, 2d, 2e, 3b, 3c and Supplementary Figs. 1, 2, 3 and 4 are uploaded on ProteinXchange and the European Nucleotide archive databases with the accession numbers PXD009403 and PRJEB28053, respectively. All other data are available from the corresponding author upon reasonable requests.

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

## Acknowledgements

This study was supported by fellowship funding to C.J.S. from the Wellcome Trust (WT105882MA). The funder played no role in the conceptualisation, design, data collection, analysis, decision to publish, or preparation of the manuscript.

## Author contributions

C.J.S., J. M. Njunge, C.A.G., J.A.B., S.B.D., J.N. and A.J.P. designed the study. C.J.S., J. M. Njunge, J. Mwongeli Ngoi, M.N.M., T.C., E.M.G., E.T.G. and A.G. conducted the experiments. C.J.S. and J. M. Njunge wrote the manuscript. All authors reviewed and approved the manuscript.

## Additional information

**Competing interests:** A.J.P. has previously conducted clinical trials of vaccines on behalf of Oxford University funded by GlaxoSmithKline Biologicals SA and ReiThera SRL but does not receive any personal payments from them. A.J.P. is the chair of the UK Department of Health (DH)'s Joint Committee on Vaccination and Immunisation (JCVI), but the views expressed in this manuscript do not necessarily represent the views of the JCVI or the DH. The remaining authors declare no competing interests.

