## [Peer Review File · Nature Communications]

Reviewers' comments:

Reviewer #1 (Remarks to the Author):

In these studies the results indicate that children infected with RSV have an increased presence of Streptococcal colonization and increased PMN-mediated anti-bacterial response based upon proteomics analysis. The studies utilize proteomics and metagenomics analyses to characterize the microbiome and presence of mediators in the airway.

While these studies nicely outline that there are clearly changes in the microbiome in the airways of children infected with RSV, they don't really link these findings to additional parameters associated with disease or immune response. Realizing that this is due to the complexity of the analysis and perhaps the number of patients that were examined, it leaves the studies underwhelming and fully descriptive without additional insight. There have been many studies that have examined bacterial infection/colonization post-RSV infection that have also identified Streptococcus and other organisms. The fact that there are more PMN in patients with either RSV and/or bacterial colonization of the airway does not move knowledge about how RSV infection alters the lung environment or impacts disease. Essentially, these data are great preliminary data to move to the next level of investigation with larger cohorts and/or more defined immune parameters to address their hypothesis that the host responses to RSV is partly directed at controlling an increased bacterial load. However, they should add data that indicates the relevance to disease as well as more depth of immune parameters.

Reviewer #2 (Remarks to the Author):

This paper builds on the existing literature that suggests that RSV infection in children triggers secondary bacterial infections, in this case focusing on streptococcus. They employ microbiome, proteome, cell culture and flow cytometry to convey to the reader that RSV infection is associated with higher burdens of streptococcus in the upper airways, which diminishes during follow up post infection. They demonstrate an antimicrobial response with upregulation of antibacterial proteins and neutrophil recruitment, confirming this functionally using inhibition assays with upper airway samples from children with RSV infection. The authors are not, however, able to conclude what activates the innate immune system, which could simply be a response to the altered microbial

profile rather than the replication of RSV. There are a number of issues I feel need addressing and I have outlined them below along with some further comments.

1) My main issue with the manuscript is that despite stating a number of times that one of the aims is “to establish associations between RSV infection and the resident airway microbiota” the authors do not present any data on the baseline difference in the microbiome between the two cohorts. Unless I am completely misreading the data presented, they are simply describing the microbiome in the cohort as a whole on Page 5 and in figure 1a. In all subsequent analysis steps there are direct comparisons drawn between the RSV positive and negative cohorts so why is this not presented for the microbiome data? Indeed in the discussion the authors themselves state on page 10 “We also found evidence that infection with RSA is associated with increased streptococcal colonisation”, without actually providing any of this data. This data should be included, even if not novel.

2) Looking at the two cohorts there appear to be differences in the oxygen saturations, between cohorts, were these statistically significant? Can this be added to the Ledger of Figure 1.

3) The authors present data showing a change in the number of streptococcus reads from baseline to convalescence (Page 6 & Figure 1c). What happens to the microbiome in patients who are RSV negative over time? Is it stable?

4) Given there were children with saturations of 57% in the longitudinal cohort were any therapies given in between sampling?

5) Were any conventional microbiological samples taken? Could the differences between cohorts simply represent the presence/absence of bacterial infection

6) In regards to the inhibition assay, the authors should state the number of subjects this was undertaken in as it is not obvious from the text, figures or legends. What percentage of each of the BII “bins” was positive and negative for RSV infection? If only the RSV + cohort are examined does the relationship between BII and streptococcus remain?

7) The flow cytometry data referred to on page 9 should be covered in more depth, as is the reader is unable to make any assessment of this data themselves. The data for the cohort as a whole should be presented. Were there any differences in cell counts between RSV + and – groups? An example of the full gating strategy should be included in the supplementary methods.

8) In the discussions on Page 10 the authors state “We hypothesised that the host response to RSV is partly directed at controlling the increased bacterial load”. However, bacterial load is not assessed at any point in the manuscript. Was any 16S qPCR work undertaken?

9) Figure 1a. There are two clear clusters on the dendrogram. Is there any statistical difference between the community structures between the two? Does the distribution of RSV + and – between the two explain this clustering?

10) The authors state that “A negative dH₂O control was incorporated into the entire experimental workflow” but then suggest they only looked at it on a Gel for evidence of contamination. Were there any sequencing controls for contamination

11) Page 10 Paragraph 1 there is a typo “airwaysi”

Responses to reviewer comments

Reviewer #1 (Remarks to the Author):

In these studies the results indicate that children infected with RSV have an increased presence of Streptococcal colonization and increased PMN-mediated anti-bacterial response based upon proteomics analysis. The studies utilize proteomics and metagenomics analyses to characterize the microbiome and presence of mediators in the airway.

While these studies nicely outline that there are clearly changes in the microbiome in the airways of children infected with RSV, they don't really link these findings to additional parameters associated with disease or immune response. Realizing that this is due to the complexity of the analysis and perhaps the number of patients that were examined, it leaves the studies underwhelming and fully descriptive without additional insight. There have been many studies that have examined bacterial infection/colonization post-RSV infection that have also identified Streptococcus and other organisms. The fact that there are more PMN in patients with either RSV and/or bacterial colonization of the airway does not move knowledge about how RSV infection alters the lung environment or impacts disease. Essentially, these data are great preliminary data to move to the next level of investigation with larger cohorts and/or more defined immune parameters to address their hypothesis that the host responses to RSV is partly directed at controlling an increased bacterial load. However, they should add data that indicates the relevance to disease as well as more depth of immune parameters.

We thank the reviewer for this comment and agree that this study forms the basis for the next level of investigations in larger paediatric cohorts. We also acknowledge the need demonstrate the relevance of the data we've presented to RSV disease pathogenesis.

To achieve this, we have undertaken a comprehensive analysis of the data, to identify whether some of the proteins that were both differentially expressed between RSV-positive and RSV-negative children and that had bactericidal properties were significantly associated with clinical measures

of RSV-disease severity. For this analysis we selected oxygen saturation as the indicator of severe RSV disease in infants since severe RSV infection typically results in impaired gas exchange leading to reduced oxygen saturation.

To understand the role of these proteins in RSV pathology, we used the median protein expression to dichotomise infants into a high protein expression group and a low protein expression group and compared oxygen saturation levels between these groups. We have generated a new figure summarising this analysis (figure 5).

The results of these analyses showed that increased expression of neutrophil granules is significantly associated with a decrease oxygen saturation measured by pulse oximetry. This association was observed for all nine proteins that were analysed. In contrast we found no association between oxygen saturation and the expression level of control proteins (beta-actin and beta-2-microglobulin).

Assuming that these results also reflect virus-host dynamics in the lower airway, they suggest that pathology of RSV disease is partly linked to neutrophil-mediated inflammation and that the influx of neutrophils into the airways (along with their associated products such as neutrophil granule proteins) contribute significantly to airway pathology and impaired gas exchange. Much of what is known about RSV-driven airway pathology has been gleaned from one histopathology study that was done using archived lung samples obtained from infants who died from suspected RSV infections in the 1930s and 1940s¹. This study reported evidence of neutrophil accumulation in the small airways and bronchioles and hinted at the role of these cells in RSV disease pathology. Our results are consistent with these observations and suggest that neutrophils may play a dominant role in the pathology of RSV.

We acknowledge that a weakness of this study is the fact the fact analysis was based on samples obtained from the upper airway. We suggest in the discussion that future studies should focus on studying the role of neutrophils in RSV pathology using lower respiratory tract samples. We are grateful to the reviewer for suggesting this new line of enquiry which we feel has substantially improved the quality of the manuscript. In addition to

this new analysis, we have also undertaken comprehensive analysis of immunological parameters in response to the second point from the reviewer. We describe this analysis at length in response to point 7 from reviewer 2.

Reviewer #2 (Remarks to the Author):

This paper builds on the existing literature that suggests that RSV infection in children triggers secondary bacterial infections, in this case focusing on streptococcus. They employ microbiome, proteome, cell culture and flow cytometry to convey to the reader that RSV infection is associated with higher burdens of streptococcus in the upper airways, which diminishes during follow up post infection. They demonstrate an antimicrobial response with upregulation of antibacterial proteins and neutrophil recruitment, confirming this functionally using inhibition assays with upper airway samples from children with RSV infection. The authors are not, however, able to conclude what activates the innate immune system, which could simply be a response to the altered microbial profile rather than the replication of RSV. There are a number of issues I feel need addressing and I have outlined them below along with some further comments.

1) My main issue with the manuscript is that despite stating a number of times that one of the aims is "to establish associations between RSV infection and the resident airway microbiota" the authors do not present any data on the baseline difference in the microbiome between the two cohorts. Unless I am completely misreading the data presented, they are simply describing the microbiome in the cohort as a whole on Page 5 and in figure 1a. In all subsequent analysis steps there are direct comparisons drawn between the RSV positive and negative cohorts so why is this not presented for the microbiome data? Indeed in the discussion the authors themselves state on page 10 "We also found evidence that infection with RSA is associated with increased streptococcal colonisation", without actually providing any of this data. This data should be included, even if not novel.

The reviewer is indeed correct that the data presented in figure 1a represents the microbiome in the entire cohort as a whole. We have now have now updated the manuscript and added a figure (1c) showing the

increased abundance of Streptococcus in RSV-infected children relative to RSV-negative children. As the reviewer states, these observations have been reported previously, but we agree on the importance of including these data here in spite of this.

2) Looking at the two cohorts there appear to be differences in the oxygen saturations, between cohorts, were these statistically significant? Can this be added to the Ledger of Figure 1.

The difference in oxygen saturation between the two groups (RSV-positive and RSV-negative) is not statistically significant (p.value =0.07). We have amended Table 1 to reflect this information by adding a column indicating P.values. We have also added a figure (supplementary figure 6) that shows the distribution of oxygen saturation values between the two groups in graphical format.

3) The authors present data showing a change in the number of streptococcus reads from baseline to convalescence (Page 6 & Figure 1c). What happens to the microbiome in patients who are RSV negative over time? Is it stable?

Unfortunately, we did not collect longitudinal samples for RSV-negative children and are unable to present that analysis in this paper. However, a recent study in children whose ages are comparable to those presented in our work² suggests that the upper airway microbiome of RSV-negative infants is not stable with time and is characterised by an increase in microbial diversity over the first few years of life.

4) Given there were children with saturations of 57% in the longitudinal cohort were any therapies give in between sampling?

The standard of care for all children admitted with saturations of <90% is administration of supplementary oxygen. This was the case for children in the longitudinal cohort as well. For these children, the acute sample was collected within an hour of admission while the convalescent sample was collected at home, approximately one month after discharge. We also

measured oxygen saturations at the convalescent time-point using fingertip pulse oximetry and as expected these measurements were all >90%

5) Were any conventional microbiological samples taken? Could the differences between cohorts simply represent the presence/absence of bacterial infection.

Yes, a blood sample was collected at admission for all children and blood culture analysis undertaken. For this study, we excluded all children who had positive blood culture results in order to avoid confounding as a result of bacteraemia. It is therefore unlikely that the results we present are simply a reflection of a difference in response between a lower respiratory virus infection and an invasive bacterial infection. We have now updated the manuscript, highlighting this exclusion criteria.

6) In regards to the inhibition assay, the authors should state the number of subjects this was undertaken in as it is not obvious from the text, figures of legends.

We are grateful to the reviewer for highlighting this omission. We have now added the number of subjects under each BII category in figure 3c.

What percentage of each of the BII "bins" was positive and negative for RSV infection?

The proportion of RSV-positive in the five BII bins was :

- a. 40% (12 RSV+, 18 RSV-negative in bin 1: BII ≤ 20)
- b. 47%, (8 RSV+, 9 RSV-negative in bin 2: BII = 40)
- c. 58%, (7 RSV+, 5 RSV-negative in bin 3: BII = 80)
- d. 57% (4 RSV+, 3 RSV-negative in bin 4: BII = 160)
- e. 100% (6 RSV+, 0 RSV-negative in bin 5: BII ≥ 320)

We have now added this information to the manuscript and have presented the data in supplementary figure 4.

If only the RSV + cohort are examined does the relationship between BII and streptococcus remain?

When only the RSV+ cohort is examined, the relationship between BII and Streptococcus remains since the abundance of Streptococcus reduces with increasing BII. This can now be seen in supplementary figure 4

7) The flow cytometry data referred to on page 9 should be covered in more depth, as is the reader is unable to make any assessment of this data themselves. The data for the cohort as a whole should be presented. Were there any differences in cell counts between RSV + and – groups? An example of the full gating strategy should be included in the supplementary methods.

We acknowledge this point and thank the reviewer for highlighting it. In response to this point as well as the last point from reviewer 1, we have made comprehensive changes in the presentation of the flow cytometry data. We have also added new data to highlight the role of airway neutrophils in the clearance of airway bacteria after an RSV infection. We have modified figure 4 to show two examples of the airway neutrophils of two infants for whom the frequency of neutrophils resident in the airway was considerably different. In addition we present a comprehensive description of the gating strategy used to identify neutrophils in the methods section and we now also show a graphical example of the gating strategy in supplementary figure 5.

In response to reviewer 1, we have added new functional data to the manuscript, in which we measure the capacity of airway neutrophils from RSV-positive and RSV-negative children to target and kill bacteria through phagocytosis. Full details of this ex-vivo flow cytometry-based functional assay are described in the methods section and sample results are shown in figure 4 (d & f). Using these additional immunological data, we compared the frequency of (i) resident airway neutrophils and (ii) the proportion of neutrophils that had phagocytosed bacteria between RSV-positive and RSV-negative children. Although the differences did not reach statistical significance, RSV-positive children in general had a higher frequency of airway neutrophils and exhibited greater capacity to phagocytose bacteria compared to RSV negative children. The failure to achieve statistical

significance is likely to be due to the small sample size used in this additional analysis. However the observed trend (higher neutrophil abundance and functional activity in RSV-positive children) is in accord with the proteomics data and further highlights the role of these cells in clearing bacterial in the wake of an RSV-infection.

8) In the discussions on Page 10 the authors state “We hypothesised that the host response to RSV is partly directed at controlling the increased bacterial load”. However, bacterial load is not assessed at any point in the manuscript. Was any 16S qPCR work undertaken?

We gratefully acknowledge this point and concede that the statement to which the reviewer refers did not accurately articulate our intended meaning. Instead we meant to say that the RSV host response is partly directed at controlling increased levels of *Streptococcus*. We are sincerely sorry for this error and have amended the manuscript to reflect our intended meaning

9) Figure 1a. There are two clear clusters on the dendrogram. Is there any statistical difference between the community structures between the two? Does the distribution of RSV + and – between the two explain this clustering?

We’ve reviewed our analysis and the distribution of RSV-positive and RSV-negative children does not appear to explain this clustering. We think this clustering would have become better aligned with the study groups had we sampled the children continuously (e.g. daily) during the course of the RSV infection. This would have allowed us to observe in a more detailed way, the delayed effects of RSV on airway microbiota as the infection progressed compared to a single acute time-point sampling. This is a consideration that we plan to adopt in our future studies .

10) The authors state that “A negative dH₂O control was incorporated into the entire experimental workflow” but then suggest they only looked at it on a Gel for evidence of contamination. Were there any sequencing controls for contamination

The negative dH2O control was included in the sequencing, and as expected, we did not obtain any reads from this control. We have updated the manuscript with this point.

11) Page 10 Paragraph 1 there is a typo "airwaysi"

We thank the reviewer for highlighting this error, which we have now corrected.

References

- 1 Johnson, J. E., Gonzales, R. A., Olson, S. J., Wright, P. F. & Graham, B. S. The histopathology of fatal untreated human respiratory syncytial virus infection. *Mod Pathol* **20**, 108-119, doi:10.1038/modpathol.3800725 (2007).
- 2 Ta, L. D. H. *et al.* Establishment of the nasal microbiota in the first 18 months of life: Correlation with early-onset rhinitis and wheezing. *The Journal of allergy and clinical immunology* **142**, 86-95, doi:10.1016/j.jaci.2018.01.032 (2018).

REVIEWERS' COMMENTS:

Reviewer #1 (Remarks to the Author):

The Authors have sufficiently addressed concerns.